# Adherence to French and ESGO Quality Indicators in Ovarian Cancer Surgery: An Ad-Hoc Analysis from the Prospective Multicentric CURSOC Study

**DOI:** 10.3390/cancers13071593

**Published:** 2021-03-30

**Authors:** Martinez Alejandra, Witold Gertych, Christophe Pomel, Gwenael Ferron, Amelie Lusque, Martina Aida Angeles, Eric Lambaudie, Roman Rouzier, Naoual Bakrin, Francois Golfier, Olivier Glehen, Michel Canis, Nicolas Bourdel, Nicolas Pouget, Pierre-Emmanuel Colombo, Frédéric Guyon, Jacques Meurette, Denis Querleu

**Affiliations:** 1Surgical Oncology Department, Institut Claudius Regaud, Institut Universitaire du Cancer—Toulouse Oncopole, 59500 Toulouse, France; ferron.gwenael@iuct-oncopole.fr (G.F.); AngelesFite.Martina@iuct-oncopole.fr (M.A.A.); 2Cancer Research Center of Toulouse (CRCT), INSERM UMR 1037, 31037 Toulouse, France; 3Obstetrics and Gynecology Department, University Hospital Lyon Sud, 69008 Lyon, France; witold.gertych@chu-lyon.fr (W.G.); francois.golfier@chu-lyon.fr (F.G.); 4Surgical Oncology Department, Centre Jean Perrin, 63000 Clermont Ferrand, France; cristophe.pomel@cjp.fr; 5Biostatistics Department, Institut Claudius Regaud, Institut Universitaire du Cancer—Toulouse Oncopole, 59500 Toulouse, France; lusque.amelie@iuct-oncopole.fr; 6Surgical Oncology Department, Institut Paoli Calmettes, 13009 Marseille, France; lambaudiee@ipc.unicancer.fr; 7Surgical Oncology Department, Institut Curie, 75248 Paris, France; roman.rouzier@curie.fr (R.R.); nicolas.pouget@curie.fr (N.P.); 8Visceral and Digestive Surgery, University Hospital of Lyon Sud, 69008 Lyon, France; naoual.bakrin@chu-lyon.fr (N.B.); olivier.glehen@chu-lyon.fr (O.G.); 9Obstetrics and Gynecology, University Hospital Clermont Ferrand, 63000 Clermont Ferrand, France; mcanis@chu-clermontferrand.fr (M.C.); nbourdel@chu-clermontferrand.fr (N.B.); 10Surgical Oncology, Institut du Cancer Montpellier, 34090 Montpellier, France; Pierre-Emmanuel.Colombo@icm.unicancer.fr; 11Surgical Oncology, Institut Bergonié, 33000 Bordeaux, France; f.guyon@bordeaux.unicancer.fr; 121203, rue des Glacis, 59500 Douai, France; jacques.meurette@assurance-maladie.fr; 13Department of Gynecologic Oncology, Agostino Gemelli University Hospital, 00168 Rome, Italy; denis.querleu@esgo.org; 14Department of Obstetrics and Gynecology, University Hospital of Strasbourg, 67091 Strasbourg, France

**Keywords:** quality indicators, ovarian cancer, surgery, France

## Abstract

**Simple Summary:**

French and European Quality Indicators have been developed for the management of ovarian cancer. In this study, we aimed to assess the ovarian cancer care distribution in France according to the volume of patients treated per hospital, and to evaluate the adherence of different centers to the quality indicators. We found that the majority of ovarian cancer patients were treated in hospitals that did not reach recommended cut-off values in terms of volume, and it is known that surgical care in low-volume hospitals is associated with worse outcome. Only 44% of high-volume centers met all the quality indicator criteria. Therefore, access to high-volume ovarian cancer providers accomplishing all the recommended institutional quality indicators is restricted to a minority of patients in France. It is mandatory that national authorities work both to improve the centralization of ovarian cancer management and to incorporate quality assurance programs into certified centers.

**Abstract:**

Background: Quality Indicators for ovarian cancer (OC) have been developed by the European Society of Gynaecological Oncology (ESGO) and by the French National Cancer Institute (Institut National du Cancer, INCa). The aim of the study was to characterize OC care distribution in France by case-volume and to prospectively evaluate the adherence of high-volume institutions to INCa/ESGO quality indicators. Methods: The cost-utility of radical surgery in ovarian cancer (CURSOC) trial is a prospective, multicenter, comparative and non-randomized study that includes patients with stage IIIC-IV epithelial OC treated in nine French health care tertiary institutions. Adherence to institutional quality indicators were anonymously assessed by an independent committee. OC care distribution in France were provided by the nationwide database of hospital procedures. Results: More than half of patients are treated in low-volume institutions. Among the nine high-volume centers participating in the study, four (44.4%) met all institutional INCa/ESGO quality indicators. The other five (55.6%) did not fulfil one of the quality indicator criteria. Conclusions: Access to high-volume OC providers in France is restricted to a minority of patients, and yet half of the referral institutions included in this study failed to meet all recommended institutional quality indicators. It is mandatory that national authorities work both to improve OC centralization and to incorporate quality assurance programs into certified centers.

## 1. Introduction

Complete removal of all macroscopic tumorous tissue is the most important prognostic factor for long-term survival in patients with advanced ovarian cancer [1]. Perioperative care, infrastructure, and especially surgical technique and experience required to safely undertake cytoreduction have improved in the last years. Rates of complete tumor resection have increased with the introduction and development of complex surgical procedures like extensive peritonectomy, en-bloc resection of the pelvis, and upper abdominal surgery. Radical procedures have increased the proportion of patients eligible for complete primary debulking with a dramatic impact on overall survival [2,3,4]. Residual disease after cytoreduction is reported to be a team/physician driven factor. There is a large amount of data on the effect of hospital and surgeon surgical volume on survival in ovarian cancer [5,6,7,8]. Surgeons who perform radical surgery in more than 50% of stage IIIc ovarian cancer cases obtain a median survival of 5.9 years compared with 2.9 years for other surgeons [9]. Mortality in ovarian cancer is also lower for patients treated in high-volume institutions, mainly due to the ability of these teams to rescue patients with complications [10]. One study using the National Cancer Database in the US showed that more than half of patients are not treated according to guidelines [6]. Quality programs are highly needed for ovarian cancer, as incomplete surgeries in non-specialized centers are a consequence of suboptimal skills and lack of knowledge on the prognostic importance of complete cytoreduction [11]. The French Government established a threshold for gynecologic cancer on 20 cases per institution per year in 2007 [12], which means that any public or private hospital can take surgically care of a single ovarian cancer case per year if the target of 20 gynecologic cancer surgeries, all sites confounded, is met. Quality indicators have been developed in France, in collaboration with the French National Cancer Institute (INCA) [11] and have also been validated by the European Society of Gynecologic Oncology (ESGO) [13]. A multidisciplinary group representing the full range of learned societies in France came to a consensus on defining quality indicators based on structural, process, and outcome indicators to assess the quality of surgical management of ovarian cancer [11]. The implementation of a quality assurance program using these indicators in France, could undoubtedly improve quality of care for ovarian cancer patients as well as quality of life and cost-effectiveness ratio.

The cost-utility of radical surgery in ovarian cancer (CURSOC) study, NCT02854215, financed by INCa (PRME 2014 Grant), aimed to provide prospective data with strong scientific evidence related to cost-utility results from compliance to quality indicators and maximal surgical effort policy, and consequently a validation of the quality assurance INCA indicators. The main objective was to investigate the potential trade-off between costs and benefits of adherence to INCA quality indicators by assessment of outcome in cost per quality-adjusted life years (QALY).

Objectives of the study were to evaluate ovarian cancer care distribution in France, to assess the proportion of institutions adherent to INCa quality indicators, and to identify factors precluding compliance to guidelines

## 2. Material and Methods

CURSOC trial is a prospective, multicenter, comparative and non-randomized study that includes consecutive patients with stage IIIC and IV epithelial ovarian, tubal, and primary peritoneal malignancies treated in nine French health care referral institutions for ovarian cancer Participating institutions are all members of the GINECO cooperative group and represent different French regions. Patients were treated according to each investigator decision based on standard of care of the investigator for patients with advanced ovarian cancer, with upfront or interval cytoreductive surgery, platinum-based chemotherapy and targeted therapies such as anti-angiogenic agents or PARP-inhibitors when appropriate. The study was approved by the national ethics committee for clinical research (Southwest and Overseas Committee for the Protection of Persons, 14 January 2016).

All participating institutions declared in a self-assessment form: number of cytoreductive surgeries per year, number of surgeons and number of cytoreductive surgeries per surgeon per year, possibility of frozen section during surgical procedure, rate of complete cytoreduction, institutional contributions to ovarian cancer clinical trials, pretreatment multidisciplinary discussion, and formalized collaboration of the surgical team with a medical oncologist, number of morbidity and mortality meetings per year. Quality indicators are presented in Appendix A.

Institutional quality indicators defined by the French Society of Gynecologic Oncology in collaboration with the INCA that were evaluated in this study [11] were:(1)Structural indicators correspond to healthcare facility resources, including equipment, number and qualification of medical staff and organizational resources, and institutional contributions to ovarian cancer trials. Indicator 1.1: A team consisting of at least two surgeons who are trained in abdominal and pelvic surgery is necessary to achieve complete cytoreduction, a minimum of ten cytoreductive surgeries per surgeon per year is required; indicator 1.2 Formalized collaboration of the surgical team with a medical oncologist; indicator 1.3 Institutional contributions to ovarian cancer clinical trials.(2)Process indicators related to patient’s management: Indicator 2.7 Possibility of frozen section at the time of surgical intervention.(3)Outcome indicators: Indicator 3.1: Rate of complete surgical resection either at initial surgery or after neoadjuvant chemotherapy, with a target of complete resection set at 70%; Indicator 3.2 Existence of a structured prospective reporting of complications within 30 days postoperatively or existence of a morbidity mortality conference.

An independent committee composed of four independent experts (DQ, EL, MC, RR) assessed adherence to INCa quality indicators to ensure homogeneity of the data analysis. Each institution was evaluated by two independent experts, unaware of the center assignment and of the other expert evaluation. In case of discordant results, a consensus was made after the analysis of the results. Participating institutions were considered as adherent to INCa quality indicators when they fulfil all structural, process, and outcome indicators that were selected for this study.

Data on distribution of ovarian cancer care in France in 2018 was extracted from the national Programme de médicalisation des systems d’information (PMSI). PMSI is a French medical-based information system that provides data on the type and volume of hospitalized diseases, and also provides the real cost of hospitalizations in France. Participation is compulsory, and it represents the French National medical database. Only cases who underwent a surgical staging or a debulking procedure were included [14].

### Statistical Analysis

Data were summarized by frequencies and percentages for qualitative variables, and by median and range (min-max) for quantitative variables. The rate of institutions adherent to INCa quality indicators was described using frequency, percentage and 95% confidence interval (CI) calculated using the exact binomial distribution. Statistical analysis was performed using the STATA software version 16 (StataCorp LLC, College Station, TX, USA).

## 3. Results

Nine institutions participated to the CURSOC trial and were included in the assessment of compliance to center quality indicators. Median number of cytoreductive procedures performed per institution per year was 50 (range 28–90). Median number of surgeons per institution was 3 (range 1–6). In 11.1% of participating institutions surgical procedures were performed by one surgeon, in 33.3% of centers by 2, in 22.2% by 3, in 11.1% by 4, 5, and 6 surgeons, respectively. In one institution, one of the surgeons performed less than 10 procedures per year. Surgical procedures were performed by a gynecologic oncologist or a trained surgeon specifically dedicated to gynecological cancer management or general surgeon specialized in peritoneal carcinomatosis in all institutions. All centers had access to frozen section analysis. All except one center achieved complete CRS in ≥70% of cases. Treatment was planned and reviewed at a multidisciplinary team meeting in all institutions. All centers participated to clinical trials in gynecologic oncology, with a median of 20% (range 11–75%) of patients to whom a clinical trial was proposed. Median number of morbidity and mortality conferences per year was 3 (range 0–12). Two centers did not have structured morbidity and mortality conferences (Table 1).

Experts classified four centers (44.4%, 95%CI: 13.7–78.8%) as adherent to INCa quality indicators. The other five centers (55.6%) did not fulfil at least one criterion: a team consisting of at least two surgeons with a minimum of ten cytoreductive surgeries per surgeon per year; rate of complete surgical resection either at initial surgery or after neoadjuvant chemotherapy, with a target of complete resection set at 70%; existence of a structured prospective reporting of complications within 30 days postoperatively or existence of a morbidity mortality conference (Table 2).

From the 530 hospitals certified for gynecologic cancer care in France, 411 surgically managed at least one case in 2018 (Table 3). The majority, 294 hospitals (71.5%) treated less than 10 cases per year, 72 hospitals (17.5%) treated between 10–19 cases, and only 45 hospitals (11%) treated at least 20 cases per year. During 2018, 3801 ovarian cancer patients underwent a surgical debulking procedure. More than half them received surgical care in low-volume institutions, and 31% in very-low volume institutions treating less than 10 cases per year (Table 4). At a geographic level, high volume institutions are concentrated in the largest French cities, with half of these institutions (18 from 45) situated in Paris, Marseille, and Lyon. From the 18 administrative regions in France, only four from the 13 located in metropolitan France have more than 3 high-volume institutions.

## 4. Discussion

This study presents data on compliance to national and European quality institutional indicators in ovarian cancer from nine high-volume cancer centers in France participating to the prospective, multicenter, comparative and non-randomized CURSOC study. More than half of the centers failed to meet all quality institution related indicators. Main reasons were complete cytoreduction rate in one center, number of cytoreductive procedures in another, and absence of morbidity and mortality rate conference in two other centers.

Rates of complete surgery vary among studies, and from upfront or interval surgical procedures, with a national cut-off for optimal rate of complete cytoreduction that has been set at 70% [11]. In our study, one of the centers had 65% of compete CRS, which is the minimum target accepted by ESGO guidelines [13]. Obtention of complete cytoreduction is dependent on surgeon’s skills and training, and on optimized infrastructure [15]. Surgeons trained for multivisceral resections and peritonectomy procedures are more likely to achieve no tumor residue at the end of surgery for advanced ovarian cancer. Indeed, radical procedures are required in more than 50% of patients, with bowel resections reported in 40%, diaphragmatic stripping in 70% and splenectomy in 25% [16]. Aletti et al. showed that a median survival rate of 2.53 years when CRS was done by surgeons performing radical surgery in less than 50% of patients, compared to a median survival of 5.9 years when CRS was performed by surgeons performing radical surgery in more than 50% of patients [9]. Although, complete resection rate could be increase with a better laparoscopic evaluation of resectability.

The ESGO has defined cutoff targets at 20 surgeries for newly diagnosed advanced ovarian cancer per year and per center and at 10 surgeries per year per surgeon [13], based on published data [5,6,7]. There is a large body of evidence from retrospective and prospective series that demonstrates the relation between hospital ovarian cancer surgical volume and likelihood to successfully render the patient with no tumor residue at the end of surgery, as well as the impact of surgical volume on OS [5,6,7]. Other studies did not find improved survival results associated to high volume hospitals and surgeons [17].

Ioaka et al. presented data on 2450 ovarian cancer cases in Japan, and found a 60% higher risk of death when patients received care in very-low-volume hospitals [8]. Bristow et al. analyzed the impact of hospital and surgeon case volume on disease-specific survival in 11,865 patients with ovarian cancer [5]. Authors reported an independent and significant 31% increase in ovarian cancer-specific survival of high case volume surgeons (≥10 cases per year) and hospitals (≥20 cases per year) compared to the low case volume pairing. They also found that only 4.3% of patients received care in high case volume surgeons and institutions [5]. This and other studies suggest geographic disparities in surgical care, with more radical procedures being performed at high-volume centers [18]. Our results also show that only 11% of hospitals with national accreditation for gynecologic cancer in France treated more than 20 cases per year. In another large series in the USA encompassing 96,802 consecutive cases, 56% of patients received care not in accordance with guidelines. Low case volume associated with non-guideline compliant care, significant higher morbidity and decreased OS. Both case-volume and adherence to treatment guidelines were independent prognostic factors for OS [6]. In the present series, one of the participating institution had a surgical team composed of five surgeons, but with only one of them performing more than 10 procedures per year. In another center, there was only one surgeon performing cytoreductive procedures. Both criterions are high impact indicators that may traduce into indirect markers of patients’ outcome. In a quality-improvement process, surgical teams from high-volume institutions or expert referral centers should also be structured to assure a minimum surgical volume per surgeon per year according to recommended cutoff. Another prospective population-based cohort showed increased survival for patients operated in teaching hospitals compared to those who received care in non-teaching centers [19]. In our study, all participating institutions were teaching hospitals.

Number of surgical procedures per year is also related to postoperative complication rate and mortality, with a 69% reduction in the risk of in-hospital death when the surgical procedure is performed by a high-volume surgeon [7]. Both national and European guidelines also highlight the importance of prospectively reporting morbidity and mortality including data on reoperations, interventional radiology, readmissions, secondary transfers to intermediate or intensive care units, and deaths [11,13]. Optimal target defined by ESGO is to prospectively record all complications among who have undergone cytoreduction for advanced ovarian cancer, and being a minimum required target the discussion of selected cases at morbidity and mortality conferences [13]. Our results show that this was the less respected indicator, with two institutions (22.2%) not meeting this minimum target, as morbidity and mortality meetings were not organized.

Length of hospital stay, complete cytoreduction rate, and hospital-related costs of care are also improved when the patient is operated in high-volume institutions [7]. Other studies demonstrate the impact of surgical specific training on gynecologic oncology in quality of life and survival [20]. Both medico-economic and quality of life impact of adherence to national quality indicators is currently being assessed in the CURSOC study.

Northern European countries, such as Denmark, the Netherlands, Sweden or Norway have centralized ovarian cancer activity with a significant increase in patient’s survival. A retrospective, nationwide, observational study in Denmark including 1160 patients with stage IIIC IV demonstrated that patients have an OS benefit from treatment in a tertiary referral center [21]. In other countries, referral centers have developed a quality assurance program with a subsequent improvement in survival [2,22]. The Mayo Clinic initiated a surgical quality program based on continuous monitoring of system structures, perioperative processes, and surgeon performance. Throughout the study period, both overall and individually, there was a higher complexity surgical score with higher rates of radical procedures reflected by a higher rated of complete procedures, with no significant increase in morbidity and mortality rates [2]. Harter et al. reported the results on therapeutic proceeding and patient outcome after the implementation of a guided interdisciplinary quality management program in 2001 in Germany. The increase in the rate of radical procedures was reflected by a proportional increase in complete CRS, from one third to two thirds, and a median survival improvement of approximately 75% [23]. It is important to highlight differences between countries in healthcare models, insurance policies, and volume centered initiatives. There can be difficulties to implement centralization associated to resistance from patients and health providers, especially in elderly patients with impaired performance status, in geographic areas with no referral institutions or in patients with poor socioeconomic status leading to increased health care disparities [24]. In our study we also demonstrated geographic disparities with half of high-volume institutions concentrated in the largest cities, and other large regions with only one high-volume center. In the Netherlands, Greving et al. showed that OC care in semi-specialized setting is a cost-effective strategy, and that centralization strategies could become cost-effective if surgical care was optimized [25]. In the USA, Bristow found that centralization was the best cost-effective strategy based on a decision-analysis model which compared estimated cytoreduction rates from a referral and a less experienced center [26]. The ESGO recommended that quality indicators be incorporated into institutional or governmental quality assurance programs in European countries. However, implementation of quality program in France is a potentially conflictual issue. Access to high-volume institutions is limited to a minority of patients. In 2018, 294 centers with national accreditation to treat gynecologic cancers treated less than 10 cases per year (71.5%). More than half of the patients in France are treated in general hospitals and private clinics who perform less than 10 cases per year, respectively, 59% and 51%, with a minority of them treated in specialized university hospitals or comprehensive cancer centers. Furthermore, only 3% of private hospitals treated more than 20 cases per year, and more than 85% of patients who underwent a debulking procedure for ovarian cancer in the private setting were treated in clinics with less than 20 cases per year. Classe et al. performed a multicenter retrospective observational cohort study in 16 French centers with accreditation to treat all gynecological cancers [27]. Authors evaluated six ESGO quality indicators: rate of complete surgical resection, participation to clinical trials in gynecologic oncology, treatment planning at a multidisciplinary tumor board, required preoperative workup, minimum required elements in operative reports, and minimum required elements in pathology reports. No institution fulfilled all indicators, and two private institutions did not meet any of them. No peritoneal score was used in 40%, no discussion in a tumor board meeting was reported in 24%, and no detail on tumor residue at the end of surgery was found in 28%. That study and this one reflect the heterogeneity of clinical care for ovarian cancer in France with large disparities between institutions, and highlights the importance of providing guidance from national societies and INCa to centralize access to high quality care. A prospective French survey representative of different types of institutions and volume is absolutely required to objectively identify national access to quality care in ovarian cancer.

Forthcoming publication of DESKTOP III trial is likely to validate CRS in management of recurrent ovarian cancer for selected patients according to AGO score. Preliminary results published at ASCO Congress [28] have already legitimated secondary CRS surgery in recurrent ovarian cancer setting performed in some tertiary French centers. It appears quite evident that secondary CRS should be considered as surgical procedure to be integrated in quality indicators. Furthermore number of secondary CRS could by lawfully integrated in quotas of surgical procedures taken into account to assess ovarian cancer centers.

The present study is noteworthy as this is the first prospective study in France to evaluate adherence of high-volume centers to institutional quality indicators defined by the INCa and the ESGO. Main limitation of this study is the self-assessment form of each institution data, that was counterbalanced by an independent committee assessment. The study may also be limited by the fact that only centers participating to the CURSOC study were evaluated, and may not reflect the results from high volume institutions in France.

## 5. Conclusions

Ovarian cancer care in high volume institutions in France is reserved to a minority of patients. More than half of the patients undergo a debulking procedure in low-volume centers performing less than 10 cases per year.

Adherence to national and European quality institutional indicators in ovarian cancer failed for at least one indicator in approximately half of the centers from nine high-volume ovarian cancer centers in France participating to the prospective CURSOC study.

Based on a large body of literature and on national and European guidelines supporting treatment in high volume institutions, INCa and French national authorities’ intervention are required to ensure delivery of quality care for all patients with ovarian cancer. First, to target patients to referral institutions and second, to incorporate quality assurance programs into certified centers.

## Figures and Tables

**Table 1 cancers-13-01593-t001:** Institution self-assessment form.

Institution Self-Assessment Form	Center 1	Center 2	Center 3	Center 4	Center 5	Center 6	Center 7	Center 8	Center 9
Number of cytoreductive procedures per year	28	40	40	47	50	60	63	75	90
Number of surgeons operating at least 10 patients a year	1	1	2	2	2	3	3	4	6
Surgeon 1: Number of cytoreductive procedures	14	40	20	23	25	20	40	30	15
Surgeon 2: Number of cytoreductive procedures	8	-	20	24	25	20	12	25	15
Surgeon 3: Number of cytoreductive procedures	3	-	-	-	-	20	11	10	15
Surgeon 4: Number of cytoreductive procedures	2	-	-	-	-	-	-	10	15
Surgeon 5: Number of cytoreductive procedures	1	-	-	-	-	-	-	-	15
Surgeon 6: Number of cytoreductive procedures	-	-	-	-	-	-	-	-	15
Percentage of patients to whom a clinical trial is proposed	20	23	75	75	20	11	15	15	20
Number of morbidity mortality meetings per year	3	6	3	0	4	3	6	12	0

**Table 2 cancers-13-01593-t002:** Expert validation of quality indicators.

Quality Indicators	Center 1 (%)	Center 2 (%)	Center 3 (%)	Center 4 (%)	Center 5 (%)	Center 6 (%)	Center 7 (%)	Center 8 (%)	Center 9 (%)
Structural indicators									
20 cytoreductive surgeries performed per center per year	Yes (100%)	Yes (100%)	Yes (100%)	Yes (100%)	Yes (100%)	Yes (100%)	Yes (100%)	Yes (100%)	Yes (100%)
At least 2 surgeons trained to achieve complete cytoreduction	Yes	No	Yes	Yes	Yes	Yes	Yes	Yes	Yes
10 surgical procedures per surgeon per year	No	Yes	Yes	Yes	Yes	Yes	Yes	Yes	Yes
Formalized collaboration of the surgical team with a medical oncologist	Yes (100%)	Yes (100%)	Yes (100%)	Yes (100%)	Yes (100%)	Yes (100%)	Yes (100%)	Yes (100%)	Yes (100%)
Institutional contributions to ovarian cancer clinical trials	Yes (100%)	Yes (100%)	Yes (100%)	Yes (100%)	Yes (100%)	Yes (100%)	Yes (100%)	Yes (100%)	Yes (100%)
Process indicators									
Possibility of frozen section at the time of surgical intervention	Yes (100%)	Yes (100%)	Yes (100%)	Yes (100%)	Yes (100%)	Yes (100%)	Yes (100%)	Yes (100%)	Yes (100%)
Outcome indicators									
Rate of complete surgical resection either at initial surgery or after neoadjuvant chemotherapy	80	95	100	100	100	100	65	90	90
Availability of a morbi-mortality report	Yes	Yes	Yes	No	Yes	Yes	Yes	Yes	No

**Table 3 cancers-13-01593-t003:** Institutions who have treated at least one ovarian cancer in 2018.

Number of Cases	Army Hospital	General Hospital	University Hospital	ESPIC *	Cancer Center	Private Hospital	Total
0	1	67	4	1	0	46	119
1 to 9	2	127	7	7	0	151	294
10 to 19	0	19	17	4	3	29	72
≥20	0	4	17	2	15	7	45
Total	3	217	45	14	18	233	530

* ESPIC: private health institution of collective interest.

**Table 4 cancers-13-01593-t004:** Distribution of patients who underwent a debulking surgical procedure in 2018 by type of institution.

Number of Cases	Army Hospital	General Hospital	University Hospital	ESPIC *	Cancer Center	Total Public Institution *N* (%)	Private Hospital *N* (%)	Total In France *N* (%)
1 to 9	9	484	37	67	0	597 (23%)	582 (51%)	1179 (31%)
10 to 19	0	241	255	45	50	591 (22%)	399 (35%)	990 (26%)
≥20	0	102	594	27	745	1468 (55%)	164 (14%)	1632 (43%)
Total	9	827	886	139	795	2656	1145	3801

* ESPIC: private health institution of collective interest.

## Data Availability

The data presented in this study are available on request from the corresponding author. The data are not publicly available due to ethical reasons.

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
