# Peer review of "Adherence to French and ESGO Quality Indicators in Ovarian Cancer Surgery: An Ad-Hoc Analysis from the Prospective Multicentric CURSOC Study"

_cancers, 2021, doi:10.3390/cancers13071593_

Round 1

Reviewer 1 Report

This is an excellent manuscript reporting on the adherence of high-volume institutions in France to national quality metrics on the management of ovarian cancer.  It highlights the challenges even renowned institutions have in meeting established guidelines and emphasizes the importance of continued quality improvement.  This manuscript adds to the growing body of literature showing that best OC outcomes are achieved in high-volume centers with experienced clinicians who follow guidelines. 

Author Response

Thank you very much for your revision and your encouraging comments, which motivate our team to keep working.

Reviewer 2 Report

  1. Please specific if ethical approval needed for CURSOC and if so, by what body it was granted.
  2. The authors note that selected quality indicators were evaluated. Why only selected? This would seem to bias the results. For transparency would provide as a supplement ALL of the quality indicator and provide more detail about these specific indicators were chosen for this particular investigation.
  3. The readers may not be familiar with PMSI - need to explain a bit more about this data system, including if participation is compulsory or voluntary, if it represents the entirety of the country, and any meaningful facets about institutional-level data entry. Additionally, are there patient-level or geographic data that may be useful in terms of deeper analyses in the current study.
  4. The authors indicate that 530 hospitals were certified for gynecologic cancer care in France, yet this study only included 9 hospitals. How were these hospitals selected? How many were eligible for inclusion in the CURSOC study, and what were the inclusion criteria? This is my biggest concern for this manuscript because the authors appear to be making sweeping conclusions based on a limited sample which may not be representative of the country as a whole.

Author Response

Dear reviewer,

Thank you for your review, we believe that your comments have improved the quality of the manuscript.

  1. Reviewer comment 1. Please specific if ethical approval needed for CURSOC and if so, by what body it was granted.

 Authors response to comment 1. CURSOC trial was granted by the French Government (PRME 2014) and received ethical approval from national Comitee (Southwest and Overseas Committee for the Protection of Persons, study approved on the 14/01/2016).Please view line 111, page 3.

  1. Reviewer comment 2. The authors note that selected quality indicators were evaluated. Why only selected? This would seem to bias the results. For transparency would provide as a supplement ALL of the quality indicator and provide more detail about these specific indicators were chosen for this particular investigation.

 Authors response to comment 2. Both French and ESGO quality indicators can be divided into institutional and patient quality indicators:

  • Institutional indicators: complete cytoreduction rate, number of procedures per center and per surgeon per year, surgery performed by a trained surgeon specifically dedicated to gynecologic cancer management, center participating in clinical trials in gynecologic oncology, treatment planned at a multidisciplinary team meeting, and existence of a prospective reporting of postoperative complications or at least discussion of selected cases at morbi-mortality meetings.
  • Patient individual indicators: required preoperative workup, preoperative, intraoperative, and postoperative management, required elements in the operative and pathology reports.

In this study, authors selected institutional indicators as the objective of the study was to assess ovarian cancer surgical care in France and the adherence of high volume centers to these QI.

To analyze all QI we would need to have access to preoperative workup, operative and pathology reports for all patients. CURSOC trial is still recruiting patients, and data on each patient participating to the study is not available.

As suggested by the reviewer, authors have included a supplementary table including all QI (Table 1. Supplementary material). 

  1. Reviewer comment 3. The readers may not be familiar with PMSI - need to explain a bit more about this data system, including if participation is compulsory or voluntary, if it represents the entirety of the country, and any meaningful facets about institutional-level data entry. Additionally, are there patient-level or geographic data that may be useful in terms of deeper analyses in the current study.

Authors response to comment 3. PMSI (Programme de médicalisation des systems d’information) is a French medical-based information system that provides data on the type and volume of hospitalized diseases, and also provides the real cost of hospitalizations in France. Participation is compulsory, and it represents the French National medical database. Explanation on the DMSI has been added to the manuscript on line 134-139, page 3. At a geographic level, high volume institutions are concentrated in the largest French cities, with half of these institutions situated in Paris, Marseille, and Lyon. Lines 188-191, page 5.

  1. Reviewer comment 4. The authors indicate that 530 hospitals were certified for gynecologic cancer care in France, yet this study only included 9 hospitals. How were these hospitals selected? How many were eligible for inclusion in the CURSOC study, and what were the inclusion criteria? This is my biggest concern for this manuscript because the authors appear to be making sweeping conclusions based on a limited sample which may not be representative of the country as a whole.

Authors response to comment 4. We also believe that this is a very important issue of the paper. CURSOC study aimed to evaluate quality indicators in high volume institutions. Even if there are 530 certified hospitals in France for treating gynecologic malignancies, there are only 45 of them surgically treating 20 or more ovarian cancers per year. Participating centers were selected by their implication in the ARCAGY GINECO French cooperative group and by their geographic location to represent different regions. Eligibility for CURSOC protocol has been further specified in the methods section, line 105-106, page 3.

Authors have also pointed out this limitation in the Discussion section, lines 332-334, page 8.

Reviewer 3 Report

In this article, the authors assessed the ovarian cancer care distribution in France according to the volume of patients treated per hospital, and evaluated the adherence of different centers to the quality indicators. 

The first issue should be considered in France, or maybe in Europe. I don’t think this is a global issue.

The centralization of ovarian cancer management is controversial. I agree that patients should undergo their treatment at a high-volume center, if possible. However, many patients with advanced-stage ovarian cancer cannot easily travel to these centers which are distant from their residence, if they have an impaired performance status, comorbidities, socioeconomic problems. Centralization to high-volume centers may not be easy in a large country such as the US [Wright, 2013 Obstet Gynecol]. In addition, Greving et al (Netherlands) concluded that current treatment of ovarian cancer patients in semi-specialized hospital settings is a cost-effective strategy, while treatment in tertiary care centers becomes only cost-effective when better surgical results would be achieved.[ 2009 Gynecol Oncol] The authors should discuss these observations, and a study by Schrag (J Natl Cancer Inst 2006) needs to be referenced.

Some of the referenced papers are not appropriate. Studies by Bristow et al. and Alletti et al. were cited, both of which evaluated effects of primary cytoreductive surgery in advanced ovarian cancer, although an indicator in this study is the rate of complete surgical resection either at initial surgery or after neoadjuvant chemotherapy (Indicator 3.1).

The objectives of this study were not stated in the Introduction section.

Author Response

Reviewer 3

Dear reviewer,

Thank you for your review, we believe that your comments will surely improve our work.

Comment 1. In this article, the authors assessed the ovarian cancer care distribution in France according to the volume of patients treated per hospital, and evaluated the adherence of different centers to the quality indicators. 

The first issue should be considered in France, or maybe in Europe. I don’t think this is a global issue.

Authors response 1. We agree with the comment of the reviewer. Ovarian cancer care distribution in France is similar to some European countries and presents important differences with other health care systems. This issue has been included in the discussion. Please view lines 283-295, page 7.

Comment 2. The centralization of ovarian cancer management is controversial. I agree that patients should undergo their treatment at a high-volume center, if possible. However, many patients with advanced-stage ovarian cancer cannot easily travel to these centers which are distant from their residence, if they have an impaired performance status, comorbidities, socioeconomic problems. Centralization to high-volume centers may not be easy in a large country such as the US [Wright, 2013 Obstet Gynecol]. In addition, Greving et al (Netherlands) concluded that current treatment of ovarian cancer patients in semi-specialized hospital settings is a cost-effective strategy, while treatment in tertiary care centers becomes only cost-effective when better surgical results would be achieved. [ 2009 Gynecol Oncol] The authors should discuss these observations, and a study by Schrag (J Natl Cancer Inst 2006) needs to be referenced.

Authors response 2. Authors have discussed on differences between countries and healthcare models, insurance policies, and volume centered initiatives. Authors have also discussed on difficulties to implement centralization associated to resistance from patients and health providers, in geographic areas with no referral institutions or in patients with poor socioeconomic status. Please view lines 283-290, page 7. As suggested by the reviewer, authors have discussed on cost-effective impact of specialized care (lines 290-295, page 7). Finally, authors have included the results from the study performed by Shrag et al (line 223-225, page 6).

Comment 3. Some of the referenced papers are not appropriate. Studies by Bristow et al. and Alletti et al. were cited, both of which evaluated effects of primary cytoreductive surgery in advanced ovarian cancer, although an indicator in this study is the rate of complete surgical resection either at initial surgery or after neoadjuvant chemotherapy (Indicator 3.1).

Authors response to comment 3. References have been updated. Bristow paper on the survival effect of complete cytoreduction has only be cited in the introduction to illustrate the importance of surgical result. Other studies published by Bristow include studies on the impact of the hospital and surgeon volume on morbi-mortality (Gynecol Oncol 2009 and Gynecol Oncol 2014), and on centralization of surgical care in ovarian cancer (Cancer 2007). Studies by Aletti cited in the manuscript evaluated the impact of the number of radical procedures per surgeon on survival (Gynecol Oncol 2006), and another study by Aletti evaluated a quality program to improve surgical care for ovarian cancer (J Am Coll Surg 2009),

Comment 4. The objectives of this study were not stated in the Introduction section.

Authors response to comment 4. The objectives of the study have been described at the end of the Introduction section. Please view lines 98-100, page 2.

Round 2

Reviewer 2 Report

All concerns have been addressed.

Reviewer 3 Report

Evaluating the current situation of ovarian cancer management is important. Also, quality indicators may be necessary to improve management for advanced ovarian cancers. However, the evaluation itself does not improve the current situation. I think that the authors are key members of the French gynecologic oncology society, so they may present more realistic future directions to national authorities and authorities in other countries in this manuscript.